# Differential Factors Are Associated with Physical Activity in Older Adults in Brazil with and without Non-Communicable Chronic Diseases: A Cross-Sectional Analysis of the 2019 National Health Survey

**DOI:** 10.3390/ijerph20146329

**Published:** 2023-07-09

**Authors:** Zainovan Serrão Pereira, Amanda Santos da Silva, João Carlos do Nascimento Melo, Jullyane Caldas dos Santos, Priscila Yukari Sewo Sampaio, Roberto Jerônimo dos Santos Silva, Raphael Henrique de Oliveira Araújo, Ricardo Aurélio Carvalho Sampaio

**Affiliations:** 1Graduate Program in Physical Education, Federal University of Sergipe, Sao Cristovao 49100-000, Brazil; zainovan38@gmail.com (Z.S.P.);; 2Department of Occupational Therapy, Federal University of Sergipe, Lagarto 49400-000, Brazil; 3Graduate Program in Health Sciences, Londrina State University, Londrina 86057-970, Brazil

**Keywords:** aging, diseases, epidemiology, Health Survey, physical activity

## Abstract

This study analyzed whether sociodemographic factors, health perception, dietary habits, and screen time are related to physical activity (PA) in older people with and without non-communicable chronic diseases (NCDs). This cross-sectional study analyzed data from the 2019 Brazilian National Health Survey; the sample was older adults (≥60 years old; n = 22,726). The outcome of this study was being physically active or inactive during leisure time, and NCD was used as a moderating variable. The correlates investigated were sociodemographic and health-related variables. According to the logistic regression analysis, it was observed that being male had an association only in the group with NCDs (OR = 1.25 (1.05–1.48)), as well as residing in the northeastern region (OR = 1.26 (1.04–1.53)). On the other hand, high levels of education (OR = 4.09 (2.92–5.2); OR = 1.92 (1.48–2.49)) and income (OR = 1.64 (1.09–2.48); OR = 1.86 (1.33–2.60)) were associated with PA in both groups, as well as dietary habits (OR = 1.03 (1.01–1.05); (OR = 1.05 (1.04–1.07)). Advanced age (OR = 0.96 (0.94–0.97); OR = 0.97 (0.96–0.98)) and reporting a regular health perception (OR = 0.53 (0.43–0.66); OR = 0.61 (0.52–0.73)) were factors associated with physical inactivity in both groups. Gender, education, and income were unequally associated with an active lifestyle in both groups, and therefore, barriers to PA may arise.

## 1. Introduction

The estimate of deaths caused annually by non-communicable chronic disease (NCD) is 41 million people, equivalent to 74% of all deaths worldwide [1]. From 2020 to 2021, the average estimate was that 75% of deaths in Brazil were caused by NCD [1]. These factors have led to the implementation of preventive measures, such as the action plan proposed by the World Health Organization (WHO), which aimed to reduce risk factors for NCD, such as tobacco consumption, salt intake, excessive alcohol consumption, physical inactivity, and insufficient physical activity [2].

Insufficient physical activity is defined as not meeting physical activity recommendations, that is, 150 min of moderate physical activity or 75 min of vigorous physical activity per week [3]. The Brazilian population has a high prevalence of insufficiently active people, about 47% in 2016, and it has been reported that 6% to 10% of mortality cases in 2008 were related to this factor [4,5,6].

Therefore, regular physical activity is associated with multiple health benefits, such as improved physical and mental function, prevention and reduction in the progression of NCD, increased years of life, and better quality of life [7,8,9,10], although these levels of physical activity tend to decrease with advancing age as it is a natural process [11].

Aging can bring functional and cognitive declines and biochemical and morphological changes in the human body [12], decreasing the levels of physical activity practice in older people [7]. Maintaining physical activity during aging protects against functional declines, decreasing the risk of premature death from chronic diseases, and there is a distinction among Brazilian population subgroups [13,14].

Access to regular physical activity practice is not equal in society, especially in developing countries [15]. People living in southern Brazil have a greater possibility of achieving physical activity recommendations compared to other regions, and these inequalities negatively affect subgroups of the Brazilian population [16].

Social factors such as sex, race, education, and economic level, in addition to physical activity, show differences and are related to the presence or aggravation of an NCD [10,15]. It is important to note that Brazil is not a homogeneous country. Thus, there are discrepancies in its population according to the region they live in [17,18]. Such differences, depending on geographic regions, refer to maintaining healthy behavior. Especially the northern, northeastern, central-western, southeastern, and southern regions show differences in this aspect [19], which can be explained by the fact that the southeastern and southern regions are the most developed regions, with better infrastructure and living conditions.

The differences observed in the literature [15,19,20] indicate that the older adults population still presents low levels of physical activity due to aging [11], which can create greater barriers for people living with NCD. Considering that these factors may remain interconnected with social inequality seen in different regions of Brazil [21], as well as access to healthcare services [19] and differences in race, education, and economic level, show significant relationships with the presence or worsening of NCD [18], causing difficulties for physical activity practice in these individuals, in addition to the limitations already present due to NCD.

Considering this, we suggest that this study can provide more reliable and updated data regarding these associations, especially considering the representative size of the national sample and the specific group studied. Therefore, it is necessary to understand factors that implicate physical activity practice [16], such as NCD in the older adults population [22], given that until now, what is known is that studies have sought to investigate certain factors in groups of people living with NCD in different populations [20]. This led our study to analyze sociodemographic factors, health perception, dietary habits, and screen time as related or not to physical activity practice in older Brazilian individuals with and without NCD through the National Health Survey (NHS) of 2019.

## 2. Materials and Methods

The present research had a cross-sectional design analyzing data from the NHS, developed in 2019 by the Brazilian Institute of Geography and Statistics (IBGE) in partnership with the Oswaldo Cruz Foundation. The NHS interviewed the population of permanent residents, grouped in census tracts in each capital, metropolitan areas, and cities, excluding indigenous villages, military bases and barracks, camps, hostels, penitentiaries, penal colonies, prisons, jails, convents, long-term care institutions, and hospitals [23].

The sample size was determined according to the 2010 demographic census conducted by the IBGE. Sampling was performed by a conglomerate, with the first stage being the census tract, followed by households within each tract, and then by one resident aged 15 years or older. Each selected household was visited by interviewers to obtain the necessary information from participants selected by simple sampling. Data collection was performed from August 2019 to March 2020. The entire process of organization and collection was coordinated by the IBGE, being the group involved composed of coordinators, supervisors, and data collection personnel. Coordinators were responsible for state units, while supervisors were responsible for advising data collection personnel (responsible for conducting the interviews). All of them were trained previously by the IBGE [23].

The NHS 2019 had a total sample of 108,525 participants, representative of the Brazilian population with different sociodemographic and geographic characteristics. In this study, only older people were selected. In this sense, 22,726 older adults of both sexes, from all Brazilian geographic areas were included in the study. The NHS does not apply specific exclusion criteria for older adults (except those living in long-term institutions and hospitals) as it aims to cover the Brazilian population across different age groups based on the most recent census. In this study, those younger than 60 years old, as well as those with missing data, were excluded from further analysis.

The outcome of the study adopted was the practice of physical activity in leisure time, using the following questions: “In the last twelve months, have you practiced any type of physical exercise or sport”? “How many days a week do you usually (used to) practice physical exercise or sport”? “In general, on the day you exercise or practice sport, how many hours does this activity last”? ‘Minutes’ “What physical exercise or sport do you practice most frequently”? The participants provided information regarding physical activity in hours and minutes; then, these data were further converted to minutes. Based on this data, the total minutes of physical activity in leisure time were calculated. Individuals were classified using the cut-off point of 150 min per week [24], and those who reported ≥150 min/physical activity/week were considered physically active in their leisure time [17,25].

The sample was divided into individuals with and without NCD, which was a moderating variable in the study. The questions used referred to the diagnosis of NCD, considering self-report of one or more diseases: heart disease, stroke, diabetes, and high blood pressure, which are the most common diseases in the Brazilian population [1,26,27]. Based on the results, individuals were dichotomized into those with and without NCD.

The independent variables investigated were: sex (female and male); age; race/color (white or non-white, where non-white refers to persons of Black, Asian, mixed-race, and Indigenous ethnicity); marital status, living with a spouse or not (with or without a spouse); and region of residence (North, Northeast, Midwest, Southeast, and South). The Southeast region was used as a reference due to its higher rate of leisure-time physical activity; education (no instruction or incomplete junior high school; complete junior high school or incomplete high school; complete high school or incomplete university; and complete university level); and per capita income, defined by four categories (up to half a minimum wage; more than half a minimum wage up to one minimum wage; more than one minimum wage up to three minimum wages; more than three minimum wages). The minimum wage in Brazil, in 2019, was R$998.00; equivalent to approximately 250 USD.

Additionally, the health-related variables analyzed were health perception, evaluated through participant self-report and categorized into three levels of “good, fair, and poor”. Eating habits correspond to the consumption of beans, vegetables, fruits, and fish. The responses ranged from 0 to 7 days for each food item (i.e., based on weekdays). The dietary score was calculated by multiplying the quantity of food consumed (portions daily) by the number of days. This multiplication resulted in a dietary score on a scale of 0 to 28, in which individuals with the highest score had good dietary habits, while those with lower scores had poor dietary habits. Television time (hours/day in front of the television); body mass index (BMI) was defined as normal weight ≤ 25 kg/m^2^; 25 < overweight < 30 kg/m^2^; obesity ≥ 30 kg/m^2^ were also considered.

### Statistical Analysis

The descriptive data were presented with a 95% confidence interval (95% CI), and a significance level of *p* ≤ 0.05 was used for the analyzed variables. For inferential statistical analysis, logistic regression was used for complex samples, considering being physically active or not in the leisure domain as the dependent (outcome) variable. The moderating variable was NCD, and the other variables were introduced as independent variables. All analyses considered sample weights and were performed using Stata 16.0.

## 3. Results

The sample of this study consisted of 22,726 older adults, with a mean age of 69.9 years (95% CI 69.7–70.0). The prevalence of physical inactivity was 81.3% (95% CI 80.4–82.2); 65.2% reported not having NCD; 56.7% were female; 50.5% reported being white; 56.3% reported not living with a spouse; 63.3% reported having no education or incomplete elementary education; 42.7% reported a monthly income between one and three minimum wages, and 46.4% lived in the southeast region.

Regarding health-related variables, 47.1% reported having “good” health; approximately 49.4% reported “normal weight” for BMI; 48.3% reported watching TV for longer periods (1–3 h/day); and 18.4% had a good score for dietary habits. The characteristics of the sample are described in Table 1.

Table 2 presents the prevalence rates of the variables divided into groups with and without NCD, through the obtained values and the overlap of the 95% CI showing that the prevalence rates are different between the analyzed groups.

The group without NCDs reported a prevalence of 82.3% (95% CI 81.3–83.4) for physical inactivity; 59.5% were female; mean age of 70.6 years (95% CI 70.4–70.8); 51.1% self-declared as non-white; 55.0% reported not living with a spouse; 47.2% reported living in the southeast region; 66.0% reported not having education or incomplete primary education; and about 42.8% reported having more than one minimum wage up to three minimum wages. For the variables related to health, 46.9% reported perceiving a “regular” state of health; 18.4% reported having healthy habits; about 46.9% reported more TV time (1–3 h/day); and 43.7% had a “normal weight” for BMI.

On the other hand, the group with NCD reported 79.4% (95% CI 77.9–80.9) of physical inactivity; 51.4% were female, with an average age of 68.5 years (95% CI 49.6–53.1); 53.6% self-declared as white; 58.6% reported living without a spouse; about 58.2% reported not having education or incomplete primary education; and 42.4% had more than one minimum wage up to three minimum wages. For the variables related to health, 61.3% reported perceiving a “good” state of health; 18.4% reported having healthy eating habits; about 50.9% reported more TV time (1–3 h/day); and 60.0% had a “normal weight” for BMI.

Table 3 shows the prevalence rates of variables divided by groups of active and inactive leisure-time physical activity. The inactive group without NCD had a prevalence of 66% (95% CI 64.9–67.1); 57.8% were female; mean age of 70.3 years (95% CI 70.1–70.5); 50.6% self-reported as non-white; 54.9% reported living with a spouse; 45.5% reported living in the southeast region; 67.8% reported not having education or incomplete elementary education; and around 42.4% reported having a monthly income of more than one up to three minimum wages. Regarding health-related variables, 44.2% reported a “fair” health status; 19.5% reported having healthy habits; about 47.2% reported longer TV time (1–3 h/day); and 49.4% had a “normal” weight according to the BMI.

On the other hand, the group of physically active older individuals without NCD had a prevalence of 61.7% (95% CI 59.2–64.2); 51.7% were female; the mean age was 67.9 years (95% CI 67.5–68.2); 55.4% self-declared as white; 62.4% reported living with a spouse; around 43.7% reported no education or incomplete elementary education; and 44% reported earning between one and three minimum wages. Regarding health-related variables, 65.5% reported a “good” health status; 19.5% reported having healthy eating habits; about 53.3% reported longer TV time (1–3 h/day); and 49.3% had “normal” weight according to BMI.

Table 4 presents the logistic regression results using the model adjusted for groups with and without NCD. In summary, the adjusted analysis showed that in the group without NCD, males had 25% higher odds of being physically active when compared to females (OR = 1.25 (1.05–1.48)); age was negatively associated with the odds of being physically active (OR = 0.97 (0.96–0.98)); people living in the Northeast region had about 26% (OR = 1.26 (1.04–1.53)) higher odds of being physically active when compared to those living in the Southeast region; those with income above three minimum wages and completed higher education had about 92% (OR = 1.92 (1.48–2.49)) and 86% (OR = 1.86 (1.33–2.60)) higher odds of being physically active, respectively; and people with a perception of “regular” health status had a higher chance of being inactive (OR = 0.61 (0.52–0.73)), while those with healthy eating habits had about 5% (OR = 1.05 (1.04–1.07)) higher odds of being physically active.

Furthermore, in the group with NCD, older age was negatively associated with physical inactivity (OR = 0.96 (0.94–0.97)); individuals with a completed higher education had approximately four times higher odds of being physically active (OR = 4.09 (2.92–5.2)); having an income higher than three minimum wages showed about 64% (OR = 1.64 (1.09–2.48)) chances of being active; having a perception of “regular” health status was negatively associated with being active (OR = 0.53 (0.43–0.66)); and those who reported having healthy eating habits had about 3% (OR = 1.03 (1.01–1.05)) higher odds of being active.

## 4. Discussion

The study aimed to analyze factors associated with leisure-time physical activity in older adults with and without NCD (i.e., sociodemographic and health-related factors). The results showed that some factors were only associated with one of the groups, such as being male and living in the Northeast region, which was only associated with the group without NCD. However, age, high levels of education, higher income, good dietary habits, and perceived health were associated with physical activity in both individuals with and without NCD.

Regarding being male and having a higher chance of being physically active in the group without NCD, similar results were found by Guedes et al. (2021) [20]. The literature has shown gender inequality in physical activity [18,28]. Several factors may explain these inequalities, such as women’s greater involvement in household tasks such as childcare, meal preparation, and cleaning [29]. Other barriers faced by women are related to the lack of support, where their partners do not encourage them to engage in physical activity, as well as the fear of violence in locations used for physical activity, barriers related to the lack of safety in certain places [30].

It is not new in the scientific literature that older people have lower levels of physical activity [7]. With advancing age, levels of physical activity tend to decrease, becoming more pronounced in people over 60 years old [31,32]. This association can be explained by reasons that vary from biological aspects such as aging to socio-environmental aspects, this factor was associated in both groups studied (with and without NCD). These limitations present with advancing age, such as declines in physiological functions, strength, balance, and cognition, are factors that contribute to individuals being insufficiently active and the aggravation of NCD [7,33]. Through this result, it is important to monitor older people to verify if they are at least reaching a minimum level of physical activity for health benefits [34].

When related to the major regions of Brazil, living in the Northeast was associated with physical activity practice in the group without NCD, considering the Southeast region as a reference. This finding differs from what the literature has been showing, since residing in the South and Southeast regions presents associations with the population’s level of physical activity [21]. The fact that we divided the sample into individuals with and without NCD in our study may explain this result, in addition to considering only older adults herein. In other studies, the sample was not divided into two groups or was not analyzed by Brazilian regions [20,35].

The sociodemographic factors, as expected, had significant associations for both groups. However, education and per capita income showed distinct values between the groups. The education level was almost four times higher in the group with NCD compared to the group without NCD. Having a high level of education and a high per capita income are significant factors for engaging in physical activity in leisure time. Guedes et al. (2021) [20] showed similar results, making it possible to analyze that the realization and encouragement of leisure-time physical activities have a higher frequency among people with higher education levels, which generates a better understanding of the importance of staying physically active, such as the necessary health care, leading these individuals to use the health system more frequently [36].

We can assume that these interactions occur due to various factors, such as socioeconomic status being a facilitator for people with NCD to seek practices as treatment for certain diseases. To alleviate these factors, the Strategic Action Plan for Coping with Chronic Non-Communicable Diseases and Disorders in Brazil was created, and in the health promotion axis, actions were established, such as the construction of squares, parks, and open areas with structures for physical activity practice [9].

The individual’s perception of health in feeling sick or healthy is not only interconnected with physical perceptions but also related to social and psychological situations [37]. It is possible to observe that not reporting a good health perception is associated with physical inactivity, and this association was the same for both groups. The lower the individual’s level of physical activity, the worse their self-perception of health will be [38]. It is important to emphasize that even among individuals with NCD, those not reporting a good health perception had a lower percentage than individuals without NCD. A better perception of health is related to maintaining physical activity [39].

Healthy eating habits were associated with the practice of physical activity in both groups with and without NCD. The Brazilian population shows the consumption of these healthy foods, thus maintaining a healthy eating pattern [40]. It is known that older people who are moderate to highly active are more likely to maintain healthy nutritional habits [41].

Despite the impact of this study, there are some limitations to point out: (i) the use of the questionnaire (NHS, 2019) to obtain information on NCD and physical activity, which were self-reported, may present memory bias; and (ii) the cross-sectional design of the study, which makes it difficult to establish causality and effect for the results found herein. Regarding the individual resident chosen, only if the person was unable to respond due to physical or mental health, another resident was selected to respond for them. Other studies based on health surveys have shown good agreement and reproducibility between the information collected and the information provided by proxy [42,43]. Using a questionnaire such as the NHS has its advantages, such as low cost, thus allowing data collection from a representative sample of the Brazilian population.

## 5. Conclusions

The results of this study showed that sociodemographic factors such as educational level and income were factors related to physical activity in both groups (with and without NCD). On the other hand, sex and region of residence remained associated with the practice of physical activity only in the group without NCD. Age and not reporting a good health perception were associated with physical inactivity in those with and without NCD. Thus, actions and public health policies for the promotion of physical activity and health, and prevention of diseases should be considered as individual and collective specificities. Finally, longitudinal studies are indicated to verify the causality of these factors in this population.

## Figures and Tables

**Table 1 ijerph-20-06329-t001:** Characterization of the variables used in the study (n = 22,726).

	Median or %	95% CI	
**Leisure-time physical activity**			
Active	18.7	17.8	19.6
Inactive	81.3	80.4	82.2
**NCD**			
With NCD	34.8	33.8	35.8
Without NCD	65.2	64.2	66.2
**Sex**			
Male	43.3	42.3	44.4
Female	56.7	55.6	57.7
**Age (years)**			
≥60 years old	69.9	69.7	70.0
**Race/color**			
White	50.5	49.4	51.6
Non-white	49.5	48.4	50.6
**Living arrangement**			
Live with partner	56.3	55.2	57.3
Live without partner	43.7	42.7	44.8
**Region**			
North	6.1	5.8	6.4
Northeast	25.4	24.5	26.2
Southeast	46.4	45.3	47.6
South	15.7	15.1	16.4
Middle-west	6.4	6.0	6.7
**Educational level**			
No instruction or <junior high school	63.3	62.2	64.3
Complete junior high school or <high school	9.5	8.9	10.2
Complete high school or <university	15.9	15.1	16.7
Complete university level	11.3	10.6	12.03
**Per capita income**			
Up to 1/2 minimum salary	10.2	9.5	10.8
>1/2 up to 1 minimum salary	31.5	30.6	32.5
+1 minimum salary up to 3 salaries	42.7	41.6	43.8
+3 minimum salaries	15.6	14.8	16.4
**Health perception**			
Good	47.1	46.0	48.1
Regular	41.7	40.7	42.8
Poor	11.1	10.6	11.9
**BMI (kg/m^2^)**			
Normal	49.4	48.3	50.5
Overweight	29.7	28.7	30.7
Obese	20.9	20.0	21.8
**TV screen time (h/day)**			
<1 h/day	22.1	21.2	23.0
≥1–3 h/day	48.3	47.2	49.4
>3 h/day	29.6	28.6	30.6
**Eating habits (score)**	18.4	18.3	18.6

Values are median or %; 95% CI, 95% confidence interval; NCD, non-communicable chronic disease; BMI, body mass index.

**Table 2 ijerph-20-06329-t002:** Characterization of the sample stratified by non-communicable diseases (n = 22,726).

	without NCD (n = 14,352)			with NCD (n = 8374)		
	Median or %	95% CI		Median or %	95% CI	
**Leisure-time physical activity**						
Active	17.7	16.6	18.7	20.6	19.1	22.1
Inactive	82.3	81.3	83.4	79.4	77.9	80.9
**Sex**						
Male	40.5	39.2	41.8	48.6	46.9	50.4
Female	59.5	58.2	60.8	51.4	49.6	53.1
**Age (years)**	70.6	70.4	70.8	68.5	68.2	68.7
**Race/color**						
White	48.9	47.5	50.2	53.6	51.9	55.4
Non-white	51.1	49.8	52.5	46.4	44.6	48.1
**Living arrangement**						
Live with partner	55.0	53.7	56.4	58.6	56.9	60.3
Live without partner	45.0	43.6	46.3	41.4	39.7	43.1
**Region**						
North	5.3	4.9	5.7	7.5	6.9	8.2
Northeast	25.1	24.1	26.2	25.8	24.4	27.1
Southeast	47.2	45.8	48.6	45.1	43.2	46.9
South	15.9	15.1	16.8	15.4	14.3	16.5
Middle-west	6.4	6.0	6.9	6.2	5.7	6.8
**Educational level**						
No instruction or <junior high school	66.0	64.7	67.3	58.2	56.4	60.0
Complete junior high school or <high school	9.6	8.8	10.4	9.4	8.3	10.4
Complete high school or <university	14.7	13.8	15.7	18.1	16.7	19.5
Complete university level	9.7	8.9	10.5	14.3	13.0	15.7
**Per capita income**						
Up to 1/2 minimum salary	9.9	9.1	10.7	10.7	9.7	11.8
>1/2 up to 1 minimum salary	33.0	31.7	34.4	28.9	27.4	30.4
+1 minimum salary up to 3 salary	42.8	41.4	44.2	42.4	40.6	44.2
+3 minimum salaries	14.3	13.4	15.3	18.0	16.5	19.4
**Health perception**						
Good	39.4	38.1	40.8	61.3	59.6	63.0
Regular	46.9	45.5	48.2	32.1	30.5	33.8
Poor	13.7	12.8	14.6	6.6	5.7	7.4
**BMI (kg/m^2^)**						
Normal	43.7	42.4	45.0	60.0	58.3	61.7
Overweight	31.3	30.0	32.6	26.0	25.2	28.3
Obese	25.0	23.8	26.2	13.2	12.0	14.5
**TV screen time (h/day)**						
<1 h/day	21.0	19.9	22.1	24.1	22.6	25.6
≥1–3 h/day	46.9	45.6	48.3	50.9	49.1	52.7
>3 h/day	32.0	30.8	33.3	25.0	23.5	26.5
**Eating habits (score)**	18.4	18.3	18.6	18.4	18.1	18.5

Values are median or %; 95% CI, 95% confidence interval; NCD, non-communicable chronic disease; BMI, body mass index.

**Table 3 ijerph-20-06329-t003:** Characterization of the sample stratified by leisure-time physical activity (n = 22,726).

	Physically Inactive (n = 18,669)			Physically Active (n = 4057)		
	Median or %	95% CI		Median or %	95% CI	
**NCD**						
Without NCD	66.0	64.9	67.1	61.7	59.2	64.2
With NCD	34.0	32.9	35.1	38.3	35.8	40.8
**Sex**						
Male	42.2	41.0	43.3	48.3	45.7	50.9
Female	57.8	56.7	59.0	51.7	49.1	54.3
**Age (years)**	70.3	70.1	70.5	67.9	67.5	68.2
**Race/color**						
White	49.4	48.2	50.6	55.4	52.8	58.0
Non-white	50.6	49.4	51.8	44.6	42.0	47.2
**Living arrangement**						
Live with partner	54.9	53.7	56.0	62.4	60.0	64.8
Live without partner	45.1	44.0	46.3	37.6	35.2	40.0
**Region**						
North	6.4	6.0	6.8	4.7	4.1	5.4
Northeast	25.9	25.0	26.8	23.0	21.1	24.9
Southeast	45.5	44.3	46.7	50.6	48.0	53.2
South	16.1	15.3	16.8	14.3	12.9	15.8
Middle-west	6.2	5.8	6.6	7.3	6.3	8.3
**Educational level**						
No instruction or <junior high school	67.8	66.6	68.9	43.7	41.1	46.3
Complete junior high school or <high school	9.2	8.5	9.9	10.9	9.3	12.5
Complete high school or <university	14.6	13.8	15.5	21.4	19.3	23.5
Complete university level	8.4	7.7	9.1	24.0	21.7	26.2
**Per capita income**						
Up to 1/2 minimum salary	11.2	10.5	12.0	5.7	4.8	6.7
>1/2 up to 1 minimum salary	33.4	32.4	34.5	23.3	21.1	25.5
+1 minimum salary up to 3 salary	42.4	41.2	43.6	44.0	41.3	46.6
+3 minimum salaries	13.0	12.2	13.8	27.0	24.8	29.2
**Health perception**						
Good	42.8	41.6	44.0	65.5	63.1	68.0
Regular	44.2	43.0	45.4	31.0	28.6	33.4
Poor	13.7	12.2	13.8	3.5	2.6	4.3
**BMI (kg/m^2^)**						
Normal	49.4	48.2	50.6	49.3	46.7	52.0
Overweight	29.2	28.2	30.3	31.8	29.4	34.2
Obese	21.4	20.4	22.4	18.8	16.8	20.9
**TV screen time (h/day)**						
<1 h/day	22.7	21.7	23.7	19.6	17.5	21.7
≥1–3 h/day	47.2	46.0	48.4	53.3	50.7	55.9
>3 h/day	30.1	29.1	31.2	27.1	24.8	29.4
**Eating habits (score)**	18.1	18.0	18.2	19.5	19.3	19.8

Values are median or %; 95% CI, 95% confidence interval; PA, physical activity; NCD, non-communicable chronic disease; BMI, body mass index.

**Table 4 ijerph-20-06329-t004:** Logistic regression considering leisure-time physical activity as a dependent variable according to non-communicable diseases in 22,721 older people.

	without NCD n = 14,351			with NCD n = 8370		
	OR	95% CI		OR	95% CI	
**Sex**						
Male	1.25	1.05	1.48	1.08	0.88	1.32
Female	Ref	Ref		Ref	Ref	
**Age (years)**	0.97	0.96	0.98	0.96	0.94	0.97
**Race/color**						
White	Ref	Ref		Ref	Ref	
Non-white	1.15	0.96	1.36	0.85	0.69	1.06
**Living arrangement**						
Live with partner	Ref	Ref		Ref	Ref	
Live without partner	0.89	0.75	1.06	1.04	0.84	1.27
**Region**						
North	1.17	0.91	1.49	0.91	0.66	1.25
Northeast	1.26	1.04	1.53	1.24	0.96	1.58
Southeast	Ref	Ref		Ref	Ref	
South	0.78	0.63	0.96	1.07	0.82	1.39
Middle-west	1.09	0.86	1.38	1.27	0.94	1.70
**Educational level**						
No instruction or <junior high school	Ref	Ref		Ref	Ref	
Complete junior high school or <high school	1.42	1.09	1.85	1.54	1.12	2.11
Complete high school or <university	1.36	1.09	1.71	1.94	1.43	2.53
Complete university level	1.92	1.48	2.49	4.09	2.92	5.62
**Per capita income**						
Up to 1/2 minimum salary	Ref	Ref		Ref	Ref	
>1/2 up to 1 minimum salary	1.36	1.02	1.80	1.32	0.92	1.88
+1 minimum salary up to 3 salary	1.54	1.16	2.04	1.42	1.00	2.08
+3 minimum salaries	1.86	1.33	2.60	1.64	1.09	2.48
**Health perception**						
Good	Ref	Ref		Ref	Ref	
Regular	0.61	0.52	0.73	0.53	0.43	0.66
Poor	0.28	0.20	0.38	0.18	0.10	0.34
**BMI (kg/m^2^)**						
Normal	Ref	Ref		Ref	Ref	
Overweight	0.93	0.78	1.12	1.10	0.89	1.35
Obese	0.86	0.70	1.05	0.92	0.68	1.24
**TV screen time (h/day)**						
<1 h/day	Ref	Ref		Ref	Ref	
≥1–3 h/day	1.17	0.95	1.44	1.06	0.84	1.34
>3 h/day	0.95	0.76	1.20	1.06	0.81	1.40
**Eating habits (score)**	1.05	1.04	1.07	1.03	1.01	1.05

Values are median or %; 95% CI, 95% confidence interval; NCD, non-communicable chronic disease; BMI, body mass index.

## Data Availability

The National Health Survey data are available at https://www.ibge.gov.br/en/statistics/social/health/16840-national-survey-of-health.html (accessed on 24 October 2022).

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
