# Peer review of "Differential Factors Are Associated with Physical Activity in Older Adults in Brazil with and without Non-Communicable Chronic Diseases: A Cross-Sectional Analysis of the 2019 National Health Survey"

_ijerph, 2023, doi:10.3390/ijerph20146329_

Round 1

Reviewer 1 Report

The pathway model that I might design for the results is confusing.  What comes first and is the causes of the causes?  is it the income that drives the presence or lack of NCD?  If so then associating NCD with PA absent an acknowledgement of intervening variables such as income education is inaccurate.   Anyone with a good education and high-income will not have a NCD and can therefore or sustain PA.  Please design a pathway model that demonstrates how these variables fit together.  

Author Response

((( Please check the attached file. )))

Thank you for your question. As long as our study has a cross-sectional design, it is difficult to establish a causal relationship. Even though, based on directed acyclic graphs (e.g., https://doi.org/10.1038/s41390-018-0071-3), we developed a pathway to show the relationship we found regarding sociodemographic characteristics, health-related variables, NCD, and physical activity at this level.

This model shows the moderator effect of NCD on the relationship among sociodemographic variables (i.e., age, sex, region, educational level, and income) and health-related variables (i.e., health perception, score of eating habits) on physical activity. We hypothesized that different factors were associated with being physically active in people with and without NCD. Although the results showed similar findings in groups (for example, education and income were related to physical activity in both people with and without NCD), they had different magnitudes. Individuals with a higher educational level, with NCD, had about four times higher odds of being physically active in reference to those with no instruction or incomplete junior high school levels.  

Reviewer 2 Report

Thank you for submitting your manuscript. After careful evaluation, I believe that this manuscript needs some minor revisions, but has the potential to be published in the future.

I have the following suggestions for the authors:

Abstract: Please consider revising the abstract after incorporating the suggested revisions throughout the manuscript.

Methods: The Methods section would benefit from more precise descriptions, particularly with regard to the created instruments and their items, as well as the study design. 

It would be beneficial for the authors to provide details regarding the timing of the study and the administration of interviews, including any preparations made for the interviewers.

Additionally, I suggest including more information about the inclusion and exclusion criteria.

Furthermore, it would be advantageous to provide more detailed information on the calculation of dietary habits and the interpretation of the various scores.

In reference to line 99-100, clarification is needed regarding the unit of measurement for the provided answers. Were they initially recorded in hours and subsequently converted into minutes, or were they directly obtained in minutes?

The abbreviation "NCD" may not be universally understood by the target audience, particularly among elderly individuals. Kindly specify whether the participants received assistance from interviewers during data collection related to this question.

Were height and weight self-reported by the participants?

Please clarify whether the questionnaire employed in this study has undergone validation procedures to establish its reliability and validity.

It is recommended that the limitations of the instruments used in the study be explicitly addressed within the limitations section of the Discussion.

Minor editing of English language required

Author Response

REVIEWER #2:

ABSTRACT

(1) Please consider revising the abstract after incorporating the suggested revisions throughout the manuscript.

A: Thank you very much; we did it.

METHODS

(2) The Methods section would benefit from more precise descriptions, particularly with regard to the created instruments and their items, as well as the study design.

A: More information were included in Methods. Although to avoid repetition, especially because our study is based on secondary data from many publications, it is recommended that readers find out detailed information on https://doi.org/10.1590/S1679-49742020000500004 and the website of the NHS. Both are well-referenced in the manuscript.

(3) It would be beneficial for the authors to provide details regarding the timing of the study and the administration of interviews, including any preparations made for the interviewers.

A: Dear reviewer, our study is based on a secondary data analysis, so the information we obtained from the NHS (Brazilian Institute of Geography and Statistics - IBGE) itself was conducted between August 2019 and March 2020. Regarding the organization of data collection and fieldwork coordination, it was carried out by the IBGE and involved coordinators, supervisors, and data collection agents. The coordinators were responsible for specific state or central units; the supervisors were in charge of data collection control and supervision of the data collection agents. The data collection agents were responsible for conducting interviews with selected residents. All individuals involved received specific training according to their respective roles.

This information was included in the manuscript.

(4) Additionally, I suggest including more information about the inclusion and exclusion criteria.

A: Dear reviewer, this is an interesting topic to be clarified. The NHS is a nationally representative study based on the most recent Brazilian census. In this sense, it aims to cover the full population, regardless of their condition, then presenting a complete panorama of them. All residents aged 15 years or older, residing in private and permanent households, are eligible for inclusion. Exclusion criteria were based on dwellings located in special or sparsely populated census sectors, such as indigenous communities, barracks, military bases, dormitories, camps, vessels, penitentiaries, penal colonies, prisons, jails, long-term care institutions for the elderly, integrated child and adolescent care networks, convents, hospitals, settlement project agrovilles, and quilombola communities. In the present study, were included people of 60 years of age or older. The exclusion criteria considered all people who did not meet the age requirement set by the study (≥ 60 years) and with missing data.

(5) Furthermore, it would be advantageous to provide more detailed information on the calculation of dietary habits and the interpretation of the various scores.

A: Dear reviewer, we selected four variables that correspond to individuals' healthy eating habits. The variables were beans, vegetables, fruits, and fish. The responses for each variable ranged from 0 to 7, corresponding to the seven days of the week. The dietary score was calculated by multiplying the quantity of food consumed (portions daily) by the number of days. This multiplication resulted in a dietary intake score ranging from 0 to 28. Therefore, individuals with higher scores had better dietary habits, while those with lower scores had poorer dietary habits. The selection of beans, vegetables, fruits, and fish was based on the Brazilian Dietary Guidelines from the Ministry of Health (available at: https://bvsms.saude.gov.br/bvs/publicacoes/dietary_guidelines_brazilian_population.pdf).

(6) In reference to line 99-100, clarification is needed regarding the unit of measurement for the provided answers. Were they initially recorded in hours and subsequently converted into minutes, or were they directly obtained in minutes?

A: The participants provided information regarding physical activity in hours and minutes; then, these data were further transformed into minutes.

(7) The abbreviation "NCD" may not be universally understood by the target audience, particularly among elderly individuals. Kindly specify whether the participants received assistance from interviewers during data collection related to this question.

A: Dear reviewer, the NHS is composed of modules, one of which refers to questions related to chronic diseases. Through this module, we selected questions regarding heart disease, stroke, diabetes, and high blood pressure, which are prevalent among the Brazilian population. For each selected disease, there was a specific question. For example, for high blood pressure, the question was, "Has any doctor ever diagnosed you with high blood pressure?" With this approach, the selected diseases correspond to a single created variable called "NCD" (Non-Communicable Diseases). Even though the participants were not asked directly about “NCD” interviewers were trained to interpret all the necessary terms accordingly to the person being interviewed.

(8) Were height and weight self-reported by the participants?

A: Height and weight data were collected by the interviewers themselves using a portable stadiometer and digital scale. The interviewers were trained in handling both types of equipment. For more details, please refer to the link: https://www.pns.icict.fiocruz.br/medidas-fisicas/

(9) Please clarify whether the questionnaire employed in this study has undergone validation procedures to establish its reliability and validity.

A: The NHS 2019 questionnaire was based on the previous NHS version (i.e., 2013). It was validated accordingly by specialists from the Brazilian Ministry of Health in terms of contents/construct, etc. The questions that in 2013 presented higher than 30% of variation coefficients were excluded in the new version. In addition, questions added to the 2019 version were tested by semantic/cognitive validation in a low socioeconomic level community. (DOI: 10.1590/S1679-49742020000500004)

(10) It is recommended that the limitations of the instruments used in the study be explicitly addressed within the limitations section of the Discussion.

A: The information regarding limitations is pointed out in the Discussion.
